# *Lactobacillus fermentum* HY7302 Improves Dry Eye Symptoms in a Mouse Model of Benzalkonium Chloride-Induced Eye Dysfunction and Human Conjunctiva Epithelial Cells

**DOI:** 10.3390/ijms241210378

**Published:** 2023-06-20

**Authors:** Kippeum Lee, Ji Woong Jeong, Jae Jung Shim, Hyun Sook Hong, Joo Yun Kim, Jung Lyoul Lee

**Affiliations:** 1R & BD Center, hy Co., Ltd., 22, Giheungdanji-ro 24 Beon-gil, Giheung-gu, Yongin-si 17086, Republic of Korea; joy4917@hanmail.net (K.L.); woongshow@hy.co.kr (J.W.J.); jjshim@hy.co.kr (J.J.S.); 2Kyung Hee Institute of Regenerative Medicine (KIRM), Medical Science Research Institute, Kyung Hee University Medical Center, Seoul 02447, Republic of Korea; hshong@khu.ac.kr

**Keywords:** conjunctiva epithelial cell, dry eye disease, *Lactobacillus fermentum* HY7302, inflammation, apoptosis

## Abstract

(1) We investigated the effects of the Lactobacillus fermentum HY7302 (HY7302) in a mouse model of benzalkonium chloride (BAC)-induced dry eye, and the possibility of using HY7302 as a food supplement for preventing dry eye. (2) The ocular surface of Balb/c mice was exposed to 0.2% BAC for 14 days to induce dry eye (*n* = 8), and the control group was treated with the same amount of saline (*n* = 8). HY7302 (1 × 10^9^ CFU/kg/day, 14 days, *n* = 8) was orally administered daily to the mice, and omega-3 (200 mg/kg/day) was used as a positive control. To understand the mechanisms by which HY7302 inhibits BAC-induced dry eye, we performed an in vitro study using a human conjunctival cell line (clone-1-5c-4). (3) The probiotic HY7302 improved the BAC-induced decreases in the corneal fluorescein score and tear break-up time. In addition, the lactic acid bacteria increased tear production and improved the detached epithelium. Moreover, HY7302 lowered the BAC-induced increases in reactive oxygen species production in a conjunctival cell line and regulated the expression of several apoptosis-related factors, including phosphorylated protein kinase B (AKT), B-cell lymphoma protein 2 (Bcl-2), and activated caspase 3. Also, HY7302 alleviated the expression of pro-inflammatory cytokines, such as interleukin-1β (IL-1β), IL-6, and IL-8, and also regulated the matrix metallopeptidase-9 production in the conjunctival cell line. (4) In this study, we showed that *L. fermentum* HY7302 helps prevent dry eye disease by regulating the expression of pro-inflammatory and apoptotic factors, and could be used as a new functional food composition to prevent dry eye disease.

## 1. Introduction

Dry eyes, also known as dry eye disease (DE), are a common disorder of the ocular surface that is characterized by an abnormal tear film caused by reduced tear production, poor tear quality, or excessive tear evaporation [1,2,3]. The characteristics of DE are usually accompanied by overall anatomic eye dysfunction, including changes in the function of the tear glands, conjunctiva, and cornea, which result in a damaged eye surface [4]. Tear dysfunction is a common problem in humans, with a reported prevalence ranging from 2 to 14.2% [5]. The increasing use of visual displays, including smartphones and personal computers, the increasing number of contact lens wearers, and the high level of fine dust in the environment contribute to the increasing prevalence of DE [6,7].

Numerous recent studies provide insight into the mechanisms of DE pathogenesis. In particular, although the etiology of DE is not fully understood, inflammation plays an important role in the development of DE [8,9]. In DE patients, inflammatory reactions are the primary cause of damage to the surface of the eye, and the elevated levels of *interleukin (IL)-1α, IL-6, IL-8*, and *tumor necrosis factor α (TNFα)* on the surface of the eye contribute to the disease pathogenesis [10,11,12,13]. Moreover, the phosphorylation of the stress-activated and mitogen-activated protein kinases (MAPKs) p38 and c-Jun *n*-terminal kinase (JNK) is associated with increased levels of pro-inflammatory cytokines [14,15]. Another contributor to ocular surface damage in DE is the overproduction of reactive oxygen species (ROS) and the resultant oxidative stress [16]. ROS is essential for regulating the physiological functions that are related to DE, and ROS production and inflammation are closely related in DE patients and animal models [16,17]. In addition, the excess cellular levels of ROS damage proteins, nucleic acids, lipids, membranes, and organelles, which can result in the activation of cell-death processes such as apoptosis [18,19,20].

Probiotics are defined as “living microorganisms that give hosts health benefits when administered in an appropriate amount” according to the definition provided by the Food and Agriculture Organization and the World Health Organization [21,22]. Probiotics include not only Lactobacillus, Bacillus, and Bifidobacterium, but also some yeast strains of Saccharomyces [23,24]. In particular, *Lactobacillus fermentum* (*L. fermentum*) belongs to the genus *Lactobacillus*, which are Gram-positive, nonsporulating, and selective anaerobic bacteria [25]. These Lactobacillus strains are recognized as safe for human consumption because they are present in fermented food, such as dairy and vegetable products. Recent studies show that *L. fermentum* has numerous biological activities such as antioxidative, antibacterial, anti-obesity, and anticancer effects, as well as beneficial effects on gut inflammation and intestinal health [26,27,28,29,30]. However, the effect of *L. fermentum* on corneal epithelial cells and alleviating DE has not been fully evaluated. *L. fermentum* HY7302 (HY7302) is a probiotic lactic acid bacterium patented in the Korean Collection for Type Cultures (KCTC 15312BP), which is isolated from Korean raw milk and is uniquely resistant to inactivation by acids, including bile acid. In this study, the anti-inflammatory and antioxidative effects of *Lactobacillus fermentum* HY7302 were analyzed in an experimental mouse model of DE and a human corneal epithelial (HCE) cell line.

## 2. Results

### 2.1. Selection of Antioxidant and Anti-Inflammatory Microorganisms from LAB

Our findings determined the antioxidative effect of the strains in the ABTS test, and the strain with the highest effect (85%) was *Lactobacillus fermentum* HY7302 (#92). When the effect of nine lactobacillus and three KCTC-type strains on IL-1β and IL-6 production in Raw264.7 cells was measured, HY7302 had the greatest inhibitory effect on cytokine production, as shown in Appendix A.

### 2.2. The Effect of HY7302 on the Corneal Epithelium and Tear Function in BAC-Induced DE Mouse Model

To investigate the effects of *L. fermentum* HY7302 in vivo, we used four groups of mice: control mice, 0.2% BAC-induced DE mice, DE plus omega-3 mice as a positive control (omega-3 is known to alleviate DE), and DE mice treated orally with 1 × 10^9^ CFU/mL/day of *L. fermentum* HY7302 (DE + HY7302) for 14 days. Corneal fluorescein staining was used to evaluate corneal epitheliopathy; Figure 1A shows representative images. The fluorescein staining score was significantly higher in the DE group than in the control group on day 14, indicating that 0.2% BAC induced corneal damage (Figure 1B). The HY7302 treatment group had a lower score than the DE group. Likewise, the TBUT and tear volume TV were significantly lower in DE mice than in control mice, and treatment with omega-3 or HY7302 prevented the BAC-induced decrease in TBUT and tear volume (Figure 1C,D).

### 2.3. The Effect of HY7302 on Corneal Epithelial Histology in the BAC-Induced DE Mouse Model

To determine the effect of HY7302 on eye morphology, H & E staining was used to assess corneal epithelial detachment in each group. Compared with the control group, the corneal epithelial structure of the DE group was irregularly shaped, as shown in Figure 2A, and this effect was reduced in the HY7302 and omega-3 groups. Moreover, the level of damage reduction was greater in the HY7302 group than in the omega-3 group. Taken together, these results indicate that HY7302 prevents the destruction of corneal epithelial cells in a mouse model of DE.

### 2.4. The Effect of HY7302 on Cell Toxicity in Human Conjunctiva Epithelial Cells

The cell viability of HCE cells (clone-1-5c-4) treated with HY7302 was evaluated using MTT and LDH assays. The results of the MTT assay showed that the cell viability was 99.21 ± 4.95, 97.87 ± 6.09, and 78.24 ± 8.56% when treated with 10^6^, 10^7^, and 10^8^ CFU/mL HY7302 (medium volume), respectively, as shown in Figure 3A. Meanwhile, the LDH assay results indicated that HY7302 was non-cytotoxic at all concentrations tested, namely, 10^6^, 10^7^, and 10^8^ CFU/mL (Figure 3C). Based on these data, subsequent in vitro studies used an HY7302 concentration of 10^6^ CFU/mL.

The effect of HY7302 on BAC-induced cytotoxicity in HCE cells was determined. As illustrated in Figure 3B,D, 0.0005% BAC was toxic to cells, as shown by a significant decrease in cell viability. However, when cells were treated with 10^6^ or 10^7^ CFU/mL HY7302, the cell viability (measured using the MTT assay) increased by 164.33% and 162.82%, respectively, compared with the BAC treatment group. In the LDH assay, cell toxicity decreased by 52.49% and 52.62% when cells were treated with 10^6^ or 10^7^ CFU/mL HY7302, respectively, compared with the BAC treatment group. Overall, these data suggest that HY7302 inhibits BAC-induced damage in conjunctiva epithelial cells.

### 2.5. The Effect of HY7302 on ROS Production in HCE Cells

Fluorescein images of DCF-DA stained cells were observed using a ZOE fluorescent cell imager system and illustrated in Figure 4. The results showed that intracellular ROS production (measured by DCF-DA fluorescence) in clone 1-5c-4 cells treated with 0.0005% BAC was 345.23% higher than the control cells, whereas ROS production in HY7302-treated cells was only 16.54% higher than the control.

### 2.6. The Anti-Apoptosis Effect of HY7302 in HCE Cells

ROS production is closely associated with apoptotic cell death [31,32]. Molecules involved in apoptosis include the transcription factors BAX and BCL2, and the sub-executor of apoptosis caspase-3 [33]. In addition, the serine/threonine protein kinase AKT plays a vital role in regulating cell survival [34]. To study whether HY7302 treatment activated apoptosis pathways, we examined the expression of Bcl-2, Bax, and caspase-3 using qRT-PCR and Western blot analyses. We found that the relative mRNA level of Bcl-2 was significantly lower in 0.0005% BAC-treated cells (0.84-fold), but HY7302 recovered the levels of Bcl-2 to levels similar to those of the control cells (1.08-fold). In addition, the mRNA level of BAX was 1.82-fold higher in BAC-treated cells than in control cells, and HY7302 treatment significantly reduced until by 1.53-fold (Figure 5A,B). As shown in the Western blotting data (Figure 5C,D), BAC treatment reduced p-PI3K/PI3K and p-AKT/AKT protein levels by 76.73% and 49.85%, respectively. When BAC-treated cells were treated with HY7302, the levels of p-PI3K/PI3K and p-AKT/AKT proteins increased slightly to 80.18% and 69.15%, respectively. In addition, the protein expression levels of the antioxidant biomarkers Bcl-2, BAX, and cleaved caspase-3 were also analyzed. In BAC-treated cells, Bcl-2 protein expression decreased to 81.67%, but it significantly increased to 128.84% when the cells were treated with HY7302. In addition, the levels of BAX protein in the BAC-treated cells were not significantly different from those in the control group (106.22%) but were significantly reduced to 86.42% by HY7302 treatment. Finally, the expression of cleaved caspase-3 was slightly higher in the BAC-treated cells than in the controls (121.37%) and was downregulated to 67.59% by HY7302 treatment.

### 2.7. The Anti-Inflammatory Effect of HY7302 in HCE Cells

After 0.0005% BAC treatment, the mRNA levels of pro-inflammatory cytokines such as IL-1β, IL-6, and IL-8 increased (Figure 6A–C) to 7.14, 15.56, and 225.25, respectively, compared with the control group. However, when cells were pretreated with 10^6^ CFU/mL HY7302, levels of IL-1β and IL-6 were significantly reduced to 4.17 and 13.13, respectively. Interestingly, the mRNA level of IL-8 increased significantly to 225.25 following BAC treatment, and the level was significantly reduced to 169.63 by HY7302 treatment.

Additionally, Western blot analysis was performed to determine the effect of HY7302 on the transcription factors and signaling pathways associated with pro-inflammation. Recent studies have shown that IL-1β activates several cellular signaling pathways, such as the MAPK phosphorylation pathway [35]. First, in BAC-treated HCE cells, the protein levels of p-ERK/ERK, p-JNK/JNK, and p-p38/p38 increased to 177.89%, 120.33%, and 121.05%, respectively. On the other hand, the levels of p-ERK/ERK, p-JNK/JNK, and p-p38/p38 proteins were recovered to 120.77%, 47.01%, and 114.58%, respectively, by HY7302 treatment. The protein expression level of IL-1β increased to 138.62% in BAC-treated cells and was reduced to 37.86% by HY7302 treatment. Finally, the protein expression level of MMP9 was increased to 898.77% by BAC treatment. MMP9 is a new diagnostic biomarker for DE, and our results showing an increase in MMP9 expression agree with published results that have led to the development of the biomarker [36,37]. HY7302 treatment significantly reduced the expression level of MMP9 by 46.02% compared with BAC-damaged cells. Taken together, our data suggest that probiotic HY7302 inhibits the synthesis of inflammatory cytokines by regulating MAPK signaling and MMP9 expression in HCE cells damaged by BAC treatment.

## 3. Discussion

LAB promote health benefits such as enhancing intestinal barrier function, inhibiting reactive oxidation stress, and modulating immune activity [38,39]. In particular, the antioxidant capacity of LAB results from the activation of molecular signaling pathways that reduce the levels of antioxidant stress enzymes in serum and tissues [40]. In addition, LAB mediate anti-inflammatory effects by reducing the levels of inflammatory cytokines, including TNFα and IL-6 [41,42]. *Lactobacillus fermentum* is a heterofermentative LAB species that is found in breast milk, dairy products, fermented plant materials, naturally fermented sausages, and saliva. In the current study, we evaluated the effects of *Lactobacillus fermentum* HY7302 using in vivo and in vitro models of DE, and studied the feasibility of using HY7302 as a food supplement for preventing DE. BALB/c mice were treated with 0.2% BAC twice a day for 14 days, and at the same time were treated with the positive control omega-3 (200 mg/kg/day) or orally administered HY7302 (1 × 10^9^ CFU/kg/day). For the in vitro study, HCE cells (clone 1-5c-4) were treated with 0.0005% BAC and 1 × 10^6^ CFU of HY7302 to determine the effect of HY7302 on the molecular signaling pathways.

We conducted screening tests on nine LAB strains isolated from raw milk obtained from Korean farms, and selected *L. fermentum* HY7302 for further study as it showed the best antioxidant and anti-inflammatory effects (Appendix A). The sequence of strain HY7302 was identified using 16S rDNA sequencing, and the sequence had 99.80% sequence similarity to closely related species, making it indistinguishable from these related species. Based on this finding, we hypothesized that HY7302 would be a candidate LAB for improving eye health, including DE.

Several studies report that BAC stimulates the apoptotic pathway via caspase-3 activation and mitochondria-dependent apoptosis. Under BAC-treated conditions, BAX is separated from the Bcl-2 complex, which allows the release of mitochondrial cytochrome c. Cytosolic cytochrome c stimulates caspase-3 cleavage to degrade cellular components [43]. In the present study, HY7302 treatment increased the expression of the pro-apoptotic protein BAX and activated caspase-3, whereas the expression of Bcl-2 slightly decreased at higher concentrations of HY7302 in human conjunctival epithelial cells. Collectively, these findings indicated that HY7302 activated an apoptotic signaling cascade involving Bcl-2, BAX, and cleaved caspase-3. As shown by the Western blot data, the Bax/Bcl2 expression ratio was higher in the HY7302 group than in the other groups. Additionally, the expression level of cleaved caspase-3 was reduced in the HY7302 group, and the p-AKT/AKT ratio was higher in the HY7302 group than in the BAC group.

Oxidative stress is known to be involved in the pathogenesis of DE. High levels of ROS cause oxidative stress and cell dysfunction, which activate cell repair mechanisms. Oxidative stress also plays an important role in ocular diseases, including ocular surface inflammation and DE [16]. In this study, we assessed the antioxidative effect of HY7302 in HCE cells damaged by BAC, and showed that HY7302 treatment reversed the BAC-induced increase in fluorescence intensity. This finding indicates that HY7302 regulates increased ROS production, preventing oxidative damage and cell dysfunction.

We investigated the molecular mechanisms of HY7302 in a mouse model of DE by focusing on the MAPK pathway. We found that HY7302 reversed the BAC increases in p-ERK/ERK, p-JNK/JNK, and p-p38/p38 expression, as well as the BAC increase in IL-1β expression. We also showed that HY7302 regulated the activity of inflammatory cytokines through the regulation of MMP9 expression in conjunctival cells, and thus exerted anti-inflammatory actions on corneal epithelial damage. MMPs are endopeptidases that degrade various components of connective tissue. In particular, MMP9 degrades gelatin, a major component of the basement membrane, and promotes the migration and infiltration of inflammatory cells during the inflammatory response [44,45]. The concentration of MMP9 in tears increases due to stress, such as inflammation or external stimuli, and this increase reduces TBUT and promotes tear evaporation. In addition, changes in MMP9 activity are one of the causes of DE syndrome; for example, MMP9 promotes the production of other inflammatory substances such as TNFα and causes inflammation of the eyelids. Therefore, MMP9 is an important biological marker for the diagnosis and treatment of DE syndrome, and inhibition of MMP9 is emerging as an effective therapeutic target for alleviating the symptoms of DE [46,47]. Several studies have shown that the consumption of probiotics can help regulate the gut microbial balance and inhibit inflammatory responses, reducing the expression of MMPs in the blood and various tissues. These findings suggest that probiotics may be effective in inhibiting inflammatory responses and preventing and treating inflammatory diseases [14,48]. However, studies on the effects of probiotics on MMP9 expression in DE are lacking. In this study, HY7302 not only effectively inhibited the BAC-induced increase in MMP9 expression in conjunctival cells but also significantly improved the TBUT in mice with DE. This result suggests that HY7801 may improve DE syndrome by regulating the expression of MMP9. However, further studies are needed to elucidate the mechanisms by which HY7032 regulates tear or ocular MMP9 levels.

Lastly, we tested the effectiveness of *L. fermentum* HY7302 in preventing BAC-induced ocular surface damage in vivo. BAC exposure caused changes in the corneal fluorescein scores and TBUT. These changes were prevented by oral administration of HY7302, and the effect of HY7302 was greater than that of omega-3. Moreover, HY7302 recovered BAC-induced decreases in the tear volume. In addition, treatment with HY7302 restored the BAC-induced increase in corneal epithelial cell separation.

## 4. Materials and Methods

### 4.1. Culture of Probiotics

Raw milk samples were collected in aseptic tubes from 24 farms nationwide in South Korea and stored at −80 °C until used in experiments. The samples were serially diluted with phosphate-buffered saline (PBS) and plated on plate count agar that contained bromocresol purple (KisanBio, Seoul, Republic of Korea). The plates were incubated for 48 h in an anaerobic chamber at 37 °C, and colonies that produced yellow rings were selected as putative lactic acid bacteria (LAB). This procedure resulted in the isolation of the probiotics *Lactobacillus fermentum* #92 (HY7302), #115, #153, #188, #189, #193, #194, #197, and #202, which were cultured on de Man, Rogosa and Sharpe broth medium (MRS; Difco, Sparks, MD, USA). The 16S rRNA sequencing was performed using universal rRNA gene primers (518F and 926R) to identify the isolated LAB strains, and processes such as gene amplification, sequencing, and database analysis were performed by Macrogen (Seoul, Korea). The strains that were identified as LAB were cultured anaerobically at 37 °C for 18 h in MRS and then stored at −80 °C until used in further experiments.

Each probiotic was cultured in a fermenter for 15–20 h at 37 °C, and then the cells were centrifuged (8000× *g*, 4 °C) for 20 min. For in vitro studies, probiotic cells were centrifuged at 2000× *g* for 10 min, washed twice with PBS, and the pellets were resuspended in PBS with pH 7.2. In addition, in screening tests, the activity of the isolated strains was compared to that of the type strains KCTC3112, KCTC3108, and KCTC3510, which were used as LAB controls.

### 4.2. Screening Test

The antioxidant enzymatic activity of the isolated strains and controls was determined using the ABTS method with an AOX-1 total antioxidant capacity assay kit (Zenbio, Durham, NC, USA). To select probiotics with antioxidant and anti-inflammatory effects, the RAW264.7 mouse macrophage cell line (KCTC) stimulated with lipopolysaccharide to induce a hyperinflammatory was used. RAW264.7 cells were cultured in Eagle’s minimum essential medium supplemented with 20% fetal calf serum and antibiotics (100 U/mL penicillin and 100 μg/mL streptomycin) in a humidified atmosphere of 5% CO_2_: 95% air at 37 °C. The relative production of IL-6 and IL-8 in LPS-stimulated RAW264.7 cells was measured using enzyme-linked immunosorbent assay (ELISA) kits (MyBioSource, San Diego, CA, USA; IL-1β MBS263843, IL-6 MBS730957) according to the manufacturer’s instructions. All samples were tested in triplicate.

### 4.3. Animal Study Design

Male BALB/c mice (ORIENT, Seongnam-si, Republic of Korea) that were 6 weeks old were used for this study. This study was approved by the Institutional Animal Care and Use Committee (IACUC number P225032) in NDIC Co., Ltd. Animals were acclimatized for at least 1 week before the study, housed under conventional conditions (12 h light/12 h dark cycle, normal appropriate humidity, and temperature under specific pathogen-free conditions), and were provided food and water ad libitum. After adaptation, the mice were randomly allocated to one of four groups (*n* = 8 per group).

To induce DE, 0.2% benzalkonium chloride (BAC, Sigma-Aldrich, St. Louis, MI, USA) dissolved in sterile PBS was applied daily to the mouse eyes at a volume of 5 μL/eye twice daily for 14 days. An equal volume of physiological saline was applied to the control group. Over the same period, HY7302 (1 × 10^9^ CFU/kg/day) or omega-3 (Ω-3, 200 mg/kg/day) in an equal volume of vehicle was orally administered daily to the mice. Omega-3 and LAB HY7302 were mixed with 0.5% aqueous carboxymethylcellulose solution. The control and DE groups were given an oral gavage of only 0.5% CMC solution.

### 4.4. Investigation of Tear Volume, Corneal Fluorescein Score and Tear Break-Up Time

The tear film destruction time (also known as TBUT) was evaluated using a cobalt blue slit lamp after the corneas were treated with 0.5% fluorescein solution on day 10 after BAC treatment. The dyed eyes were gently opened, and the time in seconds when the first crack point or the shape of the cracked line appeared on the dyed tear film layer under blue light was measured.

The tear volume and corneal fluorescein score were assessed in DE mice and controls under general anesthesia on day 14, which was induced by an abdominal injection of 10 mg/kg xylazine and 100 mg/kg ketamine. The amounts of tears were measured using Schirmer’s test strips (Bio Color Tear Test, Bio Optics, Seongnam-si, Republic of Korea), which were cut to 1/2 thickness.

When the TBUT evaluation was complete, one drop of an antidote (troperin, Korean Alcon, Seoul, Republic of Korea) was injected into the cornea and the excess antidote was removed. Then, 0.2% fluorescein sodium salt was applied to the cornea. After lightly washing with saline, the corneal image was recorded under blue light (Micron-IV, Phoenix, Kawasaki, Japan), and the corneal area was divided into five areas according to the National Eye Institute guidelines to score 0–3 points per area, and the values were added to calculate a combined score.

### 4.5. Hematoxylin and Eosin (H&E) Staining

To determine corneal epitaxial defects, corneal samples from the mouse eyes were fixed with 4% paraformaldehyde at room temperature for 24 h. The tissues were then paraffin-embedded, and the resulting paraffin blocks were cut into 5 µm sections and stained with H & E to assess the histology.

### 4.6. Culture of Human Chang Conjunctiva Cells, Clone 1-5c-4

Human conjunctiva epithelial (HCE) cells, clone 1-5c-4, were cultured in DMEM/F12 supplemented with 5% fetal bovine serum (Gibco, Thermo Fisher Scientific, Waltham, MA, USA), 1X penicillin/streptomycin. All cells were cultured in 5% CO_2_ at 37 °C in a humidified CO_2_ culture incubator. All chemicals were dissolved in DMSO (Sigma-Aldrich, USA). HY7302 was prepared in MRS broth (BD Difco, Detroit, MI, USA) and added to cells at a final concentration of 10^6^ CFU/5 × 10^5^ cells, 10^7^ CFU/5 × 10^5^ cells, and 10^8^ CFU/5 × 10^5^ cells. Cultured HCE cells were treated with 0.0005% (*v*/*v*) BAC to induce dry eye conditions.

### 4.7. Cell Toxicity Test

HCE cells were seeded (5 × 10^4^ cells/well) in 96-well plates and incubated overnight in a culture medium. Cells were then treated with HY7302 culture medium (10^6^, 10^7^, and 10^8^ CFU/mL) and incubated for a further 24 h. Next, 0.5 mg/mL 3-(4,5-dimethyl-2-thiazolyl)-2,5-diphenyl-2H-tetrazolium bromide (MTT) solution was added and incubated for 4 h. The MTT-containing medium was removed, and 200 μL of DMSO was added to elute formazan crystals. The absorbances of the eluates were measured at 595 nm on an ELISA microreader (BioTek, Winooski, VT, USA).

Lactate dehydrogenase (LDH) activity was measured as an indicator of cytotoxicity. HCE cells were seeded at a density of 1 × 10^5^ cells per well in 48-well plates in a culture medium. The cells were then treated with various concentrations (10^6^, 10^7^, and 10^8^ CFU/mL) of HY7302 for 24 h. Cytoplasmic LDH was measured in the cell-free supernatant from HY7302-treated cells using an optimized LDH test (G1780, Promega, Madison, WI, USA) according to the manufacturer’s instructions. The percentage of viable cells was calculated as 100% by defining the cell viability without treatment.

### 4.8. Quantitative Real-Time Polymer Chain Reaction (qRT-PCR)

RNA was isolated from cells using an easy-spin total RNA extraction kit (iNtRON Biotechnology, Seongnam-si, Korea) and cDNA was obtained using a thermal cycler (Bio-Rad, Hercules, CA, USA) using a Maxime RT PreMix (iNtRON Biotechnology). Then, the cDNA was analyzed by qRT-PCR (Applied Biosystems, Carlsbad, CA, USA) using the TaqMan probe-based gene expression analysis system in combination with TaqMan gene expression master mix (Applied Biosystems, Waltham, MA, USA). The primers used in the experiment were as follows: B-cell lymphoma protein-2 (Bcl-2, Hs04986394_s1), B-cell lymphoma protein 2-associated X (BAX, Hs00180269_m1), IL-1β (IL-1β, Hs01555410_m1), IL-6 (IL-6, Hs00174131_m1), IL-8 (IL-8, Hs01337546_g1), and glyceraldehyde 3-phosphate dehydrogenase (GAPDH, Hs02786624_g1). Expression data were normalized to GAPDH. The mRNA levels were calculated as a ratio using the 2^−ΔΔCT^ method for comparing groups of data generated by qRT-PCR.

### 4.9. Western Blots

Tissues were lysed by pro-prep buffer (iNtRON Biotechnology) containing proteinase inhibitors and phosphatase inhibitors. The protein quantity was checked using a protein assay kit (Bio-Rad). Protein samples (18 µg) were resolved on sodium dodecyl sulfate-polyacrylamide gel electrophoresis (SDS-PAGE) gels and then transferred to activated polyvinylidene fluoride (PVDF) membranes. Primary antibodies purchased from Cell Signaling Technology (Danvers, MA, USA) against the following proteins were used: PI3 Kinase p110α C73F8 (PI3K, 4249), protein kinase B (AKT, 4685), phospho-AKT D9E Ser473 (p-AKT, 4060), B-cell lymphoma 2 D55G8 (Bcl-2, 4223), Bcl-2 gene family E4U1V (BAX, 41162), cleaved caspase-3 Asp175 (cleaved cas-3, 9664), c-JNK (8252), phospho-JNK (p-JNK, 9255), p38 MAPK D13E1 (p38, 54470), phospho-p38 MAPK D3F9 Thr180/Tyr182 (p-p38, 4511), IL-1β (D3U3E, 12703), matrix metallopeptidase-9 D6O3H (MMP9, 13667), and glyceraldehyde 3-phosphate dehydrogenase D16H11 (GAPDH, 5174). The following antibodies were purchased from Invitrogen, Carlsbad, CA, USA: Phospho-PI3K Thyr458/Tyr199 (p-PI3K PA5-118549), extracellular signal-regulated kinase1/2 P77-MAPK (EKR, PA13-8600), and phospho-ERK1/2 P44-MAPK Thr202/Tyr204 (p-EKR, PA602-310). The membranes were blocked with 5% non-fat dried milk for 3 h and then incubated with a secondary antibody conjugated to IgG horseradish peroxidase.

### 4.10. Evaluation of ROS Production

ROS production was measured using a dichlorofluorescin diacetate (DCF-DA) assay. The HCE cells were seeded at a density of 5 × 10^5^ cells per well in 24-well plates in a culture medium. The cells were then pretreated with 10^6^ CFU/mL of HY7302 for 24 h. For the in vitro DE model, cells were treated with 0.0005% BAC for 3 h. Then, 50 μM of the cell-permeable fluorogenic probe DCF-DA was added to each cell well for 20 min. DCF-DA that remained in the medium was removed and cells were washed twice with PBS buffer. Intercellular ROS intensity levels and DCFH-DA fluorescence (exitation485/emission535) were visualized and measured using an ELISA microreader (BioTek) to determine the DCF generated through cellular oxidation.

### 4.11. Statistical Analysis

All statistical results were presented as mean ± standard error of means. Data were compared statistically with unpaired two-tailed Student’s t-test using SPSS version 26.0 (IBM, Somers, NY, USA). In the case of in vivo study, the control and DE group were presented by *p* value < 0.05 (^#^), < 0.01 (^##^), and < 0.001 (^###^). Significant differences between HY7302 group and DE were presented by *p* value < 0.05 (*), < 0.01 (**), and < 0.001 (***). In the case of in vitro study, the control cells and BAC-treated group were presented by *p* value < 0.05 (#), < 0.01(##), and < 0.001(###). Significant differences between HY7302-treated cells and BAC-treated cells were presented by *p* value < 0.05 (*), < 0.01 (**), and < 0.001 (***).

## 5. Conclusions

In summary, we illustrated the efficacy of *L. fermentum* HY7302 as a therapeutic supplement for preventing DED. Our selected probiotic regulates not only ROS-dependent BAX/Bcl-2 and caspase-3 pathway-mediated apoptosis but also MMP9-mediated pro-inflammatory cytokine production in corneal epithelial cells. To further analyze and prove the mechanism of HY7302 in DE, future studies should analyze the effects of HY7302 on intestinal microbiome changes. Such studies will enable us to identify the interactions and links between intestinal microbial populations and eye health.

## Figures and Tables

**Figure 1 ijms-24-10378-f001:**
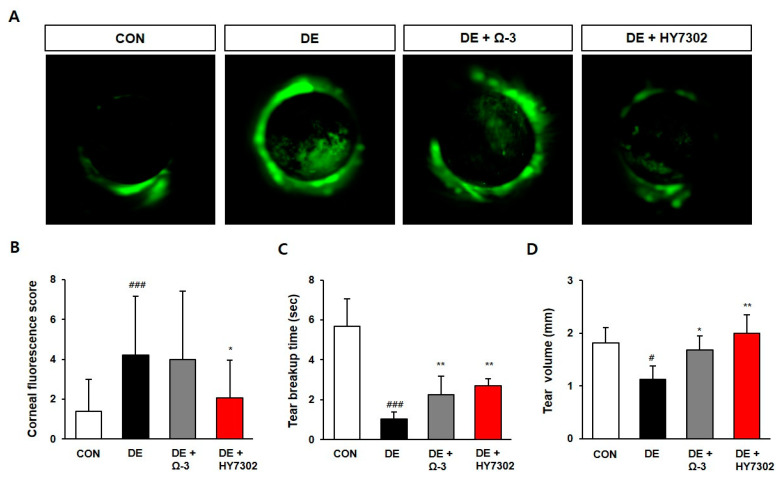
(**A**) Representative images of fluorescein-stained cornea and corneal fluorescein scores on day 14. (**B**) The corneal fluorescein scores were measured according to the National Eye Institute guidelines for dry eyes (DE) based on the images photographed using a Micron-IV ophthalmic imaging system on Day 14. (**C**) The tear break-up time was measured using a fluorescein strip. (**D**) The tear volume was measured by Schirmer’s test on Day 14. The groups were assigned as follows: CON (vehicle); DE (0.2% BAC treatment); DE + Ω-3 (0.2% BAC treatment with oral omega-3 administration); DE + HY7302 (0.2% BAC treatment with 10^9^ CFU/mL HY7302). A total of four groups of mice (*n* = 7) were studied. The control and DE group were presented by *p* value < 0.05 (#) and < 0.001 (###). Significant differences between Ω-3 and HY7302 groups and DE were presented by *p* value < 0.05 (*) and < 0.01 (**).

**Figure 2 ijms-24-10378-f002:**
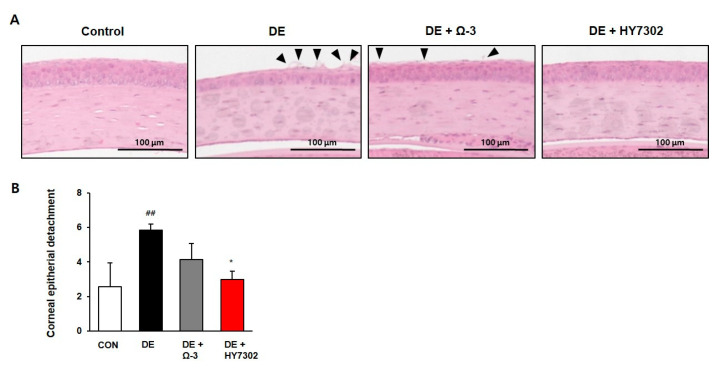
The effect of omega-3 and HY7302 on the detachment of corneal epithelial cells. (**A**) Representative hematoxylin and eosin images of mouse cornea tissue sections are presented. The arrows indicate detached apical cells. Original magnification, ×40; (**B**) The numbers of detached corneal epithelial cells are expressed as the mean ± the SD (*n* = 3–4 eyes). Each group was assigned as follows: CON (vehicle); DE (0.2% BAC treatment); DE + Ω-3 (0.2% BAC treatment with oral omega-3 administration); DE + HY7302 (0.2% BAC treatment with 10^9^ CFU/mL HY7302). The control and DE group were presented by *p* value < 0.01 (##). Significant differences between HY7302 groups and DE were presented by *p* value < 0.05 (*).

**Figure 3 ijms-24-10378-f003:**
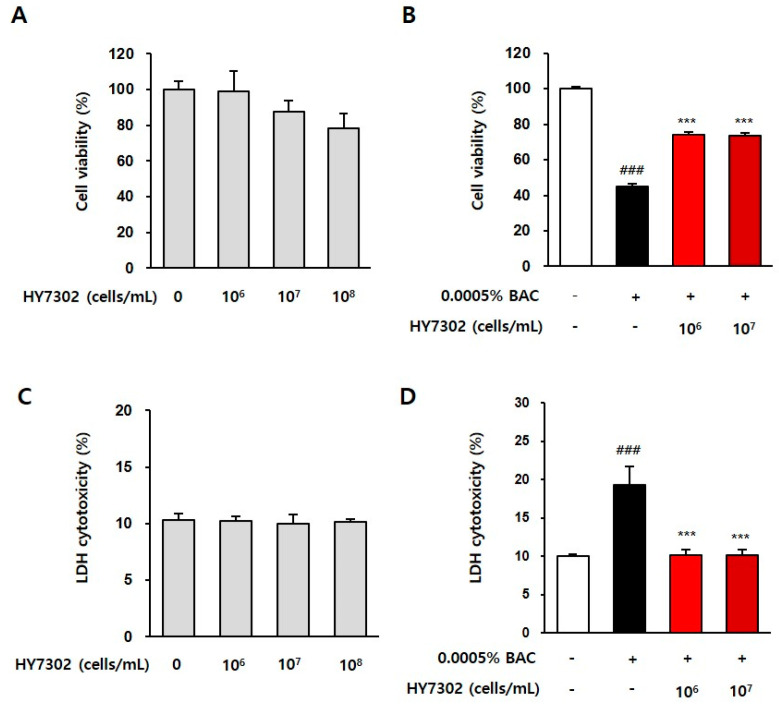
The cell toxicity of Lactobacillus fermentum HY7302 on the clone-1-5c-4 cell line. (**A**,**B**) The 3-(4,5-dimethylthiazol-2-yl)-2,5-diphenyltetrazolium bromide test; and (**C**,**D**) the lactate dehydrogenase (LDH) release cytotoxic assay. Normal cells or cells exposed to 0.0005% benzalkonium chloride (BAC) for 4 h were treated with 10^6^, 10^7^, and 10^8^ CFU/mL HY7302. The results are shown as a percentage relative to the control samples. The control and DE group were presented by *p* value < 0.001 (###). Significant differences between Ω-3 and HY7302 groups and DE were presented by *p* value < 0.001 (***).

**Figure 4 ijms-24-10378-f004:**
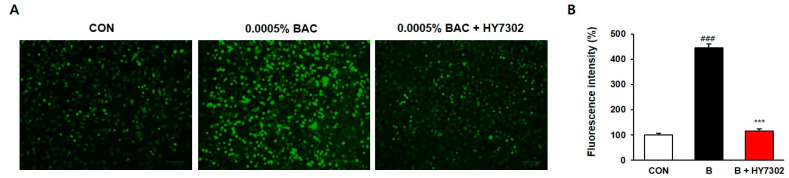
Dichloro-dihydro-fluorescein diacetate (DCF-DA) staining data. (**A**) The confocal fluorescence microscopy observations in 0.0005% BAC exposed clone-1-5c-4 cell line with or without HY7302 (original magnification 400×). (**B**) The relative fluorescence intensity is shown as a percent normalized to negative-control cells. Cells were treated with *Lactobacillus fermentum* HY7302 at a concentration of 10^6^ CFU/mL. The control cell and BAC-treated cells were presented by *p* value < 0.001 (###). Significant differences between HY7302 and BAC were presented by *p* value < 0.001 (***).

**Figure 5 ijms-24-10378-f005:**
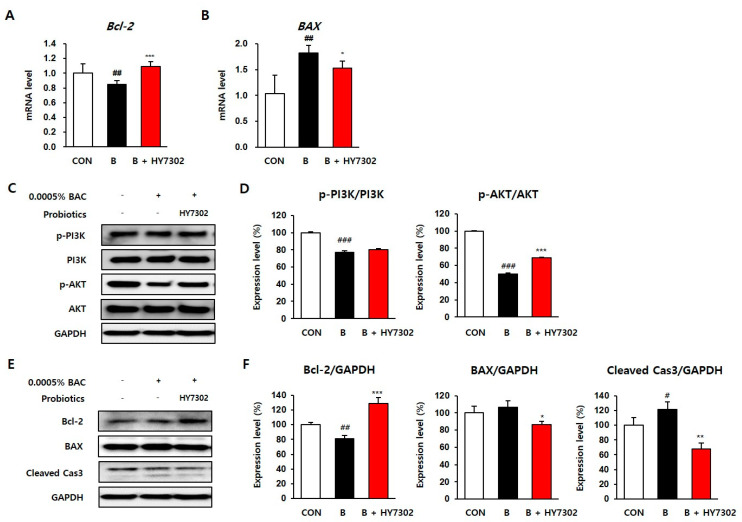
The effect of *Lactobacillus fermentum* HY7302 on PI3K/AKT pathway and apoptotic factors in 0.0005% BAC-treated clone-1-5c-4 cell line. (**A**,**B**) The mRNA levels of *B-cell lymphoma 2* (*Bcl-2*) and *Bcl-2-associated X protein* (*BAX*) in cells were normalized to the level of glyceraldehyde 3-phosphate dehydrogenase (GAPDH) mRNA and calculated as a relative-fold value; (**C**,**D**) Western blot analysis of phosphoinositide 3-kinase (PI3K), phospho-PI3K (p-PI3K), protein kinase B (AKT), phospho-AKT (p-AKT), GAPDH, and quantified relative protein expression; (**E**,**F**) Western blot analysis of Bcl-2, BAX, cleaved caspase-3, and quantified relative protein expression. Statistical significance was measured using one-way ANOVA followed by Tukey’s post hoc test (*n* = 4). Datasets denoted by different letters are significantly different; *p* < 0.05. CON, control cells; B, 0.0005% BAC-treated cells; B + HY7302, BAC-treated cells treated with 10^6^ CFU/mL of *Lactobacillus fermentum* HY7302 The control cell and BAC-treated cells were presented by *p* value < 0.05 (#), < 0.01 (##), and < 0.001 (###). Significant differences between HY7302 and BAC were presented by *p* value < 0.05 (*), < 0.01 (**) and <0.001 (***).

**Figure 6 ijms-24-10378-f006:**
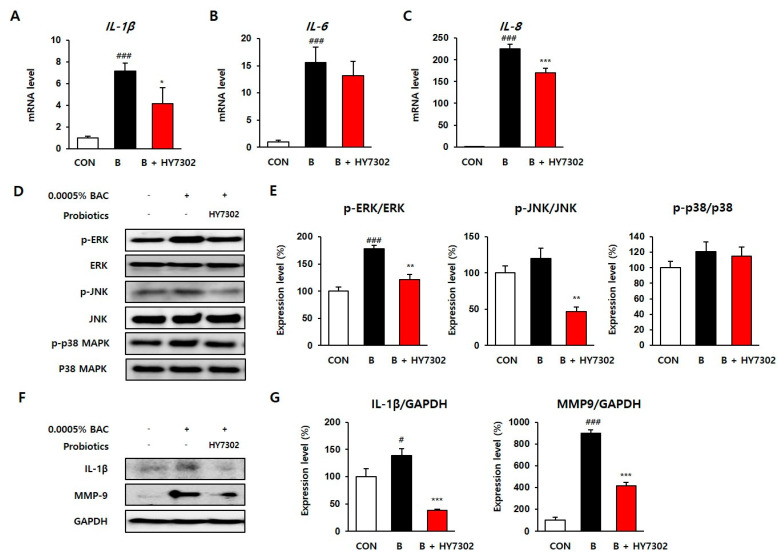
The effect of *Lactobacillus fermentum* HY7302 on pro-inflammatory factors in 0.0005% BAC-treated clone-1-5c-4 cell line. (**A**–**C**) The mRNA levels of *interleukin-1 beta* (*IL-1β*), *IL-6*, and *IL-8* in cells were normalized to the level of GAPDH mRNA and calculated as a relative-fold value. (**D**,**E**) Western blot analysis of extracellular signal-regulated kinases (ERK), phospho-ERK (p-ERK), c-Jun *N*-terminal kinase 1 (JNK), phospho-JNK (p-JNK), mitogen-activated protein kinase (p38 MAPK), phospho-p38 MAPK (p-p38 MAPK) GAPDH, and quantified relative protein expression. (**F**,**G**) Levels of IL-1β, metalloproteinase-9 (MMP9), and GAPDH in cells. Statistical significance was measured using one-way ANOVA followed by Tukey’s post hoc test (*n* = 4). Datasets denoted by different letters are significantly different; *p* < 0.05. CON, control cells; B, 0.0005% BAC-treated cells; B + HY7302, BAC-treated cells treated with 10^6^ CFU/mL of *Lactobacillus fermentum* HY7302 The control cell and BAC-treated cells were presented by *p* value < 0.05 (#) and < 0.001 (###). Significant differences between HY7302 and BAC were presented by *p* value < 0.05 (*), < 0.01 (**) and <0.001 (***).

## Data Availability

The data presented in this study are available.

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
