# Peer review of "Lactobacillus fermentum HY7302 Improves Dry Eye Symptoms in a Mouse Model of Benzalkonium Chloride-Induced Eye Dysfunction and Human Conjunctiva Epithelial Cells"

_ijms, 2023, doi:10.3390/ijms241210378_

Round 1

Reviewer 1 Report

The experimental design is good. Here are a few comments:

Line 30 - need revision as - "L. fermentum HY7302  helps prevent dry eye disease by 30 regulating the expression of pro-inflammatory and apoptotic factors, and could be used as a new 31 functional food composition to prevent dry eye disease."

Figure 2 A - for DE - difficult to see the detached apical cells.

In Figure 5 C and Figure 6D,  is there any positive control used along with targeted markers like GAPDH? 

Clinical studies or human sample data will be precious to look at these effects. Is there any data for Lactobacillus fermentum HY7302 use in humans for therapeutic purposes? If yes, would like to see the information share here. 

Needs improvement. 

Author Response

Dear reviewer 1

Thank you for considering our manuscript for publication in International Journal of Molecular Sciences. We are very pleasure to have been given the opportunity to revise our manuscript, and appreciate for your insights. We have added content for better understanding; “Lactobacillus fermentum HY7302 Improves Dry Eye Symptoms in a Mouse Model of Benzalkonium Chloride-Induced-Eye Dysfunction and Human Conjunctiva Epithelial Cells”.

We hope that the manuscript is now acceptable for publication in International Journal of Molecular Sciences and declare that authors of this work have no conflict of interests.

Sincerely,

Jung-Lyoul Lee, Ph.D

Review 1

The experimental design is good. Here are a few comments:

1.Line 30 - need revision as - "L. fermentum HY7302 helps prevent dry eye disease by 30 regulating the expression of pro-inflammatory and apoptotic factors, and could be used as a new 31 functional food composition to prevent dry eye disease."

=>Thank you for your kind advice. We revised it.

2.Figure 2 A - for DE - difficult to see the detached apical cells.

=>Thanks for the advice. We replaced the pictures with areas where epithelial cell detachment was more clearly observed in the DE group and resized all pictures. In addition, we attached H&E retaining data of DE mouse model induced by 0.2% BAC treatment. The data of this reference paper and our DE image are similar (Please see the detached image in fire).

(Kim, Kyung-A., et al. "The leaves of Diospyros kaki exert beneficial effects on a benzalkonium chloride–induced murine dry eye model." Molecular Vision 22 (2016): 284.)

3.In Figure 5 C and Figure 6D, is there any positive control used along with targeted markers like GAPDH?

=>Thank you for your advice. According to the reviewer's advice, GAPDH was added to Figures 5C and 6D to further improve understanding, and the expression amount of phosphorylated form compared to the total form was calculated.

4.Clinical studies or human sample data will be precious to look at these effects. Is there any data for Lactobacillus fermentum HY7302 use in humans for therapeutic purposes? If yes, would like to see the information share here.

=> We agree with the reviewer's advice and believe that clinical trials as a further study must be conducted. In addition, it is important to investigate the relationship between intestinal cells and dry eyes in animal models before clinical trials, so we are planning an in vivo study related to this. In particular, we will investigate the correlation with MMP9 in eye tissue related to microbiome bacterial total changes and eye dryness using the mic model.

Reviewer 2 Report

The authors have evaluated the effects of Lactobacillus in reversing the effects of BAC. The study design is good. I have minor concerns-

1. If authors could provide a schematic showing the possible mechanisms of bac that were revered with Lactobacillus or its mode of action on the corneal surface.

2. Please specify why you used lactobacillus and not other probiotics. 

3. Another point in discussion that can be added is about BAC preservatives used in AGM, where probiotics can be helpful.

The authors can run the manuscript in English editing software for minor corrections

Author Response

Dear reviewer 2

Thank you for considering our manuscript for publication in International Journal of Molecular Sciences. We are very pleasure to have been given the opportunity to revise our manuscript, and appreciate for your insights. We have added content for better understanding; “Lactobacillus fermentum HY7302 Improves Dry Eye Symptoms in a Mouse Model of Benzalkonium Chloride-Induced-Eye Dysfunction and Human Conjunctiva Epithelial Cells”.

We hope that the manuscript is now acceptable for publication in International Journal of Molecular Sciences and declare that authors of this work have no conflict of interests.

Sincerely,

Jung-Lyoul Lee, Ph.D

Review 2

The authors have evaluated the effects of Lactobacillus in reversing the effects of BAC. The study design is good. I have minor concerns-

1.If authors could provide a schematic showing the possible mechanisms of bac that were revered with Lactobacillus or its mode of action on the corneal surface.

=>We sincerely appreciate your advice. We are currently conducting further research on the mechanism involved in dry eye and eye cell protection by probiotics. Briefly, we are studying the effects of changes in the intestinal microflora following intake of probiotics and related metabolites on the oxidative damage and inflammatory response of the cornea and conjunctiva. We will reflect your advice and sketch it when we write a manuscript for our next study.

2.Please specify why you used lactobacillus and not other probiotics.

=>Commonly known probiotics are lactobacillus and bifidobacterium. Also, some lactococcus and bacillus species are known as probiotics. Among them, lactobacillus is a fungus that is found in various plant and animal-derived fermented foods and has been consumed by mankind for a long time. Probiotics derived from foods with a long history of consumption can be considered advantageous in terms of safety. Bifidobacterium species are also known to have useful functionality as probiotics, but most of them are derived from the intestines and feces of animals, so additional research is essential to verify safety for ingestion. In this study, only probiotics derived from food raw materials were used as comparison targets, and lactobacillus is predominantly found in probiotics commonly detected in food. HY7302 is a lactobacillus strain derived from raw milk. We will consider the effects of probiotics other than lactobacillus on ocular tissues when conducting further research according to your advice. In particular, we will investigate the correlation with MMP9 in eye tissue related to microbiome bacterial total changes and eye dryness using the mic model. Thanks again for the advice.

3.Another point in discussion that can be added is about BAC preservatives used in AGM, where probiotics can be helpful.

=> Thank you very much for the reviewer's advice. It is very interesting that BAC, which Latobacillus fermentum HY7302 is used in anti-glaucoma media (AGM), can be beneficial for eye health in relation to preservatives. However, since our experiments are mainly the efficacy of probiotic oral-administration in mice models, we believe that it is unreasonable to include the effects of preservatives directly treated in the eyes. This is well worth considering later through further research.

Reviewer 3 Report

In this manuscript, the author demonstrated that Lactobacillus fermentum HY7302 Improved dry eye symptoms in a mouse model of benzalkonium chloride-induced-eye dysfunction and prevented dry eye disease by regulating the expression of pro-inflammatory and apoptotic factors in vitro. In my opinion, some issues should be further addressed, and I hope the following comments could be helpful in improving the paper.

1.     Please carefully follow the formatting requirements in the author's guide. The results and discussion section of this article are above.

2.     In the abstract, the description of the results is incomplete, for example, the description of p-ERK/ERK, p-JNK/JNK, and p-p38/p38 proteins are missing.

3.     Line 118, Please add the 0.2% BAC dosage volume and frequency.

4.     Line 122, carboxymethylcellulose should be abbreviated CMC for its first appearance.

5.     Anesthesia of mice will affect the determination of the tear volume, corneal fluorescein score, and tear break-up time (TBUT). These indicators should be measured without anesthesia.

6.     The subheading content is missing in 3.1.

7.     There are many errors in the text, such as”987.87±6.09” (Line 267), “0005%” (Line 289), “All statistical results were presented as mean ± standard error of means ± SEM.”(Line 211).

8.     “Based on these data, subsequent in vitro studies used a HY7302 concentration of 106 CFU/mL.”(Line 270), Cells were treated with Lactobacillus fermentum HY7302 at a concentration of 108 CFU/mL(Figure 3), “However, when cells were pretreated with 107 CFU/mL HY7302,”(Line 333). The dosage of HY7302 in this paper is very confusing.

9.     What is the connection between ROS and fluorescence intensity?

10.  Figure 5 shows pi3k, which is not described in the article.

11.  In Figure 6, there is no significance in p-JNK/JNK, and p-p38/p38. so the analysis of them in the paper is biased.

Author Response

Dear reviewer 3

Thank you for considering our manuscript for publication in International Journal of Molecular Sciences. We are very pleasure to have been given the opportunity to revise our manuscript, and appreciate for your insights. We have added content for better understanding; “Lactobacillus fermentum HY7302 Improves Dry Eye Symptoms in a Mouse Model of Benzalkonium Chloride-Induced-Eye Dysfunction and Human Conjunctiva Epithelial Cells”.

We hope that the manuscript is now acceptable for publication in International Journal of Molecular Sciences and declare that authors of this work have no conflict of interests.

Sincerely,

Jung-Lyoul Lee, Ph.D

Review 3

In this manuscript, the author demonstrated that Lactobacillus fermentum HY7302 Improved dry eye symptoms in a mouse model of benzalkonium chloride-induced-eye dysfunction and prevented dry eye disease by regulating the expression of pro-inflammatory and apoptotic factors in vitro. In my opinion, some issues should be further addressed, and I hope the following comments could be helpful in improving the paper.

1.Please carefully follow the formatting requirements in the author's guide. The results and discussion section of this article are above.

=>Thank you for your kind advice. We revised it.

2.In the abstract, the description of the results is incomplete, for example, the description of p-ERK/ERK, p-JNK/JNK, and p-p38/p38 proteins are missing. p-ERK/ERK, p-JNK/JNK, p-p38/p38.

=>We added to Abstract that the expression of p-ERK/ERK, p-JNK/JNK, and p-38/p38 up-regulated by BAC was significantly reduced by HY7302 treatment.

3.Line 118, Please add the 0.2% BAC dosage volume and frequency.

=> We revised as “To induce DE, 0.2% benzalkonium chloride (BAC, Sigma-Aldrich, St. Louis, MI, USA) dissolved in sterile PBS was applied daily to the mouse eyes at a volume of 5 μL/eye twice daily for 14 days.” in Line 121.

4.Line 122, carboxymethylcellulose should be abbreviated CMC for its first appearance.

=> We rewritten carboxymethylcellulose as the abbreviation CMC.

5.Anesthesia of mice will affect the determination of the tear volume, corneal fluorescein score, and tear break-up time (TBUT). These indicators should be measured without anesthesia.

=> Thank you for pointing out our mistake. The mouse TBUT test was performed first on day 10 of BAC processing. On the following 14 days, tear volume and corneal phosphor score tests were performed with anesthesia. We re-wrote this in the method.

6.The subheading content is missing in 3.1.

=>The subheading content of result 3.1 was written as " Selection of antioxidant and anti-inflammatory microorganisms from LAB.". Line 224

7.There are many errors in the text, such as”987.87±6.09” (Line 267), “0005%” (Line 289), “All statistical results were presented as mean ± standard error of means ± SEM.”(Line 211).

=>Thank you for your kind advice. We revised it.

8.“Based on these data, subsequent in vitro studies used a HY7302 concentration of 106 CFU/mL.”(Line 270), Cells were treated with Lactobacillus fermentum HY7302 at a concentration of 108 CFU/mL(Figure 3), “However, when cells were pretreated with 107 CFU/mL HY7302,”(Line 333). The dosage of HY7302 in this paper is very confusing.

=>Thank you for pointing out our mistake. The concentration of the Lactobacillus fermentum HY7302 was used in vivo studies was109 CFU/mL. On the other hand, Cell were treated with HY7302 at a concentration of 106 CFU/mL in vitro studies.

9.What is the connection between ROS and fluorescence intensity?

=>As written in Method 2.10., the intercellular ROS intensity levels, visualized by DCFH-DA fluorescence, were measured on a BioTek Synergy H1 hybrid microplate reader (BioTek, Winooski, VT, USA) with excitation and emission wavelengths of 485 nm and 530 nm, respectively.

10.Figure 5 shows pi3k, which is not described in the article.

=> We revised the duplicate use of p-AKT/AKT to p-PI3K/PI3K.

11.In Figure 6, there is no significance in p-JNK/JNK, and p-p38/p38. so the analysis of them in the paper is biased.

=> The asterisks were missed in p-JNK/JNK and p-p38/p38 in Figure 6, and significance is indicated now.
